# Performance Improvement of Dual-Pulse Heterodyne Distributed Acoustic Sensor for Sound Detection

**DOI:** 10.3390/s20040999

**Published:** 2020-02-13

**Authors:** Xiangge He, Min Zhang, Lijuan Gu, Shangran Xie, Fei Liu, Hailong Lu

**Affiliations:** 1College of Engineering, Peking University, Beijing 100871, China; hexiangge@pku.edu.cn (X.H.); hlu@pku.edu.cn (H.L.); 2Beijing International Center for Gas Hydrate, Peking University, Beijing 100871, China; zhang_min@pku.edu.cn (M.Z.); liufei19900213@126.com (F.L.); 3Institute of Ocean Research, Peking University, Beijing 100871, China; 4Max Planck Institute for the Science of Light, Staudtstr. 2, 91058 Erlangen, Germany; shangran.xie@mpl.mpg.de

**Keywords:** distributed acoustic sensor, Rayleigh backscattering, phase fading, sound detection

## Abstract

Phase fading is fatal to the performance of distributed acoustic sensors (DASs) influencing its capability of distributed measurement as well as its noise level. Here, we report the experimental observation of a strong negative correlation between the relative power spectrum density (PSD) at the heterodyne frequency and the noise floor of the detected phase for the heterodyne demodulated distributed acoustic sensor (HD-DAS) system. We further propose a weighted-channel stack algorithm (WCSA) to alleviate the phase fading noise via an enhancement of the relative PSD at the heterodyne frequency. Experimental results show that the phase noise of the demodulated signal can be suppressed by 13.7 dB under optimal condition. As a potential application, we exploited the improved HD-DAS system to retrieve a piece of music lasted for 205 s, demonstrating the reliability of detecting wideband sound signal without distortion.

## 1. Introduction

Acting as the sensing element, optical fiber has several inherent advantages including immunity to electromagnetic radiation, high flexibility, low cost and so on. As a result, distributed optical fiber sensors, with the optical fiber functioning as both sensing and transmitting medium, have attracted great attention from scientific and industrial communities [1,2]. Since distributed optical fiber sensor is based on light scattering originated from fluctuations in the physical properties of the sensing fiber, it can be categorized into three groups based on Raman scattering [1], Brillouin scattering [3], and Rayleigh scattering [4], respectively. Among all those approaches, distributed acoustic sensor (DAS) exploiting coherent Rayleigh backscattering to acquire the acoustic-wave-induced strain variation on the fiber has been extensively explored over the last decade [5,6].

Since DAS can obtain the dynamic acoustic field along the sensing fiber with length of several kilometers, it is ideal for applications like structural health monitoring [7,8], intrusion detection [9], leak detection [10], geological sensing [11,12], and oilfield exploration [13]. Especially when applied in the oilfield, with harsh environment of high temperature and pressure, one can adopt the pre-installed optical fiber or optimize the optical fiber to meet special requirements. In this scenario, field tests of DAS applied in vertical seismic profile, micro-seismic monitoring, hydraulic fracturing monitoring, and reservoir surveillance have been reported [14,15]. Jousset et. al. recently used the DAS system to interrogate the pre-installed communication optical fiber cables to record seismic signals from natural and man-made sources on Reykjanes Peninsula, SW-Iceland, shedding a light on earth hazard assessment and exploration [16].

The detected signal of DAS comes from coherent interference between Rayleigh backscattered light from different positions along the fiber. Since the amplitude and the phase shift of the backscattered signal are randomly distributed over distance, ripple like fluctuations appear on top of the interference intensity. When the backscattered signal superimposes destructively, the detected signal is vanishingly small, which is known as the intensity fading. The fading effect is fatal in the DAS system because, at the fading positions, the acoustic signal is lost. To overcome this issue, Hartog et al. used multiple optical interrogation frequencies with diverse fading properties to aggregate the data obtained to substantially reduce the fading noise [17]. However, this scheme greatly increases the system complexity, especially at the source part. Chen et al. used an intensity modulator in their time-gated digital optical frequency domain reflectometry configuration based DAS system. The harmonics induced by the intensity modulator were fully adopted to suppress fading noise via using matched filter and rotated-vector-sum method [18].

Different from the previously mentioned single pulse configurations, we recently proposed a dual-pulse heterodyne demodulated DAS (HD-DAS) approach that can retrieve the acoustic-induced phase modulation in a distributed manner [19]. In this approach, a pulse pair with fixed frequency difference functions as a moving interferometer along the sensing fiber, and the phase information can be demodulated with a heterodyne detection algorithm. An extremely low noise floor and a high signal-to-noise ratio (SNR) can be achieved in this system due to the fact that the pulse pair share the noise properties when propagating along the same fiber. The phase fading effect was also characterized in the HD-DAS system and was found to be dominated by the random phase retardant rather than the scattering coefficient [20].

Here, in this work, we experimentally investigate the phase fading effect and propose a new approach on improving the HD-DAS system by alleviating the phase fading effect. Through a detailed signal analysis of the collected signal, it is found that, along the sensing fiber, the relative power spectrum density (PSD) at heterodyne frequency has a negative correlation relationship with the noise floor of the detected phase. In addition, the root mean square (RMS) of the phase noise increases sharply when the relative PSD at heterodyne frequency is lower than 50 dB. Based on this observation, we proposed a weighted-channel stack algorithm (WCSA) to suppress the noise floor through the enhancement of the relative PSD at heterodyne frequency. Finally, we use the improved HD-DAS system to recover the acoustic signal induced by a piece of music demonstrating the effectiveness of the proposed approach.

## 2. Working Principle of the HD-DAS System

Figure 1 illustrates the working principle of HD-DAS system, in which a heterodyne pulse pair, offset in both temporal and frequency domains, are injected into the sensing optical fiber. The heterodyne pulse pair functions similar to the sensing and reference arms of an interferometer. Thus, the phase change induced by the external acoustic field is modulated onto the heterodyne frequency, which can be retrieved using a heterodyne demodulation algorithm [19]. As the two pulses propagate along the same fiber, the noise properties are shared with each other and thus self-cancelled. Therefore, this system can retrieve the demodulated signal with high signal-to-noise ratio (SNR) after heterodyne detection.

To numerically simulate the DAS system, the discrete model of Rayleigh backscattering is commonly adopted [21]. In this model, the optical fiber is considered as a series of discrete reflectors with a length much longer than the optical wavelength. The reflectivity of each reflector follows the Rayleigh distribution and the reflected phase is uniformly distributed in [−π,π]. The electric field of the backscattered pulse pair from position *z* can be expressed as [20]
(1)$$\begin{array}{cc} E(z,\tau ) & =\left(E_{1}+\delta E_{1}\right)\int_{z}^{z+\frac{w}{2}}r\left(p,\tau \right)e^{j\theta (p,\tau )}exp\left[j2\int_{0}^{p}\phi \left(l,\tau \right)dl\right]e^{-j\omega_{1}\tau +j\delta \phi_{1}}dp\\ \; & +\left(E_{2}+\delta E_{2}\right)\int_{z+\frac{L_{d}}{2}}^{z+\frac{L_{d}+w}{2}}r\left(p,\tau \right)e^{j\theta (p,\tau )}exp\left[j2\int_{0}^{p}\phi \left(l,\tau \right)dl\right]e^{-j\omega_{2}\tau +j\delta \phi_{2}}dp \end{array}$$
where $\tau =1/f_{r}$ is the sampling time within each analog to digital converter (ADC) channel, and $f_{r}$ is repetition rate of the pulse pair. $E_{i}\left(i=1,2\right)$ is the field amplitudes of each pulse in the pulse pair with amplitude fluctuation of $\delta E_{i}$. The pulse pair are modulated to frequency of $f_{1}$ and $f_{2}$ with an identical pulse width of *w*. The frequency offset, i.e., the heterodyne frequency is given by $\Delta f=f_{1}-f_{2}$, while the spatial distance between them is $L_{d}$. $\omega_{i}=2\pi f_{i}\left(i=1,2\right)$ indicates their angular frequency. r(p,τ) and θ(p,τ) are the reflectivity and phase of the reflector. ϕ(l,τ) is the acoustic-induced phase variation at position *l* and time τ. The phase noise generated with pulse pair propagation are represented by $\delta \phi_{1}$ and $\delta \phi_{2}$. As shown in Equation (Equation 1), the detected signal has a complicated expression resulting from the interference between backscattered lights from different reflectors along the fiber. Since the scattering amplitude and phase shift of those Rayleigh scatters are randomly distributed along the sensing fiber, the collected interference signal appears as intensity fluctuations, which is known as fading effect.

Figure 2 shows the configuration of our HD-DAS system. A narrow linewidth, approximately 100 Hz, CW laser at central wavelength of 1550.12 nm (NKT Koheras BasiK E15, Denmark) is used as the light source. The narrow linewidth laser was used to suppress the laser phase noise. The output of the laser is divided into two optical paths by an optical coupler (OC1). Then, two acousto-optic modulators, AOM1 with frequency of 100 MHz and AOM2 with frequency of 100.05 MHz, are used to generate the heterodyne pulses with a pulse width of *w* = 6 m and a repetition rate of $f_{r}$ = 200 kHz. The heterodyne frequency is Δf = 50 kHz. A $L_{d}$ = 10 m delay fiber is placed after AOM2 to temporally offset the two pulses. The heterodyne pulse pair is amplified by an Erbium-doped fiber amplifier (EDFA1) before injecting into the sensing fiber through the optical fiber circulator. The Rayleigh backscattered signal is boosted by another EDFA before sending it into the photo detector (PD). Then, a high speed data acquisition card (DAQ) samples the data with a rate of 100 MS/s. Finally, the data are transmitted to a computer to implement the heterodyne demodulation algorithm [19].

## 3. Results and Discussions

### 3.1. The Phase Fading Phenomenon in the HD-DAS System

The 500 m sensing fiber is isolated in a box and the waterfall plot of the demodulated phase signal under steady condition is shown in Figure 3b, along with the root mean square (RMS) value of the retrieved phase at different positions plotting in Figure 3a. The standard deviation of the RMS is ∼0.72 rad. We notice that the retrieved phase signal fluctuates randomly along the sensing fiber and also varies with time. Note that the common mode noises in the system (including laser intensity and phase noises, AOM modulation noise and environmental noise) partially contribute to the phase noise, while they are not the key reason for the phase fading since they are identical to all sampling channels. The inhomogeneity of the phase noise mainly results from the phase fading noise [20] inherent to the DAS system. In addition, the phase fading noises also vary with time indicating that the parameter of *r* and θ in Equation (Equation 1) are functions of time. This further causes the position of the fading channels fluctuates over time.

Using a 100 MS/s ADC sampling frequency, the sampling length for each channel is 1 m and the entire 500 m long optical fiber contains 500 sampling channels. Two typical channels (channel 163 with low noise and channel 314 with high noise) with distinct noise levels are therefore chosen to be analyzed and compared.

Figure 4 summarizes the results from channel 163, in which Figure 4a is the collected intensity signal and Figure 4b is its PSD. The 1 s time duration is chosen as a compromise between negligible time-dependent effect of RMS and the sufficient frequency resolution to resolve the PSD value at the heterodyne frequency. The red dashed line in Figure 4b marks the noise floor (estimated as the averaged PSD value from 1 kHz to 49 kHz) and the red diamond marks the PSD value at the heterodyne frequency Δf, i.e., 50 kHz. The resultant relative PSD at the heterodyne frequency is $\Delta PSD_{\Delta f}$ = 80.4 dB defined as the difference between the PSD at Δf (−12.2 dB V${}^{2}$/Hz) and the noise floor (−92.6 dB V${}^{2}$/Hz). Figure 4c plots the demodulated phase signal along with its PSD shown in Figure 4d. The corresponding noise floor, calculated as the averaged PSD value from 100 Hz to 20 kHz, is −83.5 dB rad${}^{2}$/Hz. The visible peak at 10 kHz (as well as the harmonics in Figure 4b) is caused by the unavoidable switching effect in the AOM pulse modulator.

As a comparison, Figure 5 plots results for channel 314. In this case, the noise floor of the intensity signal is −98.8 dB V${}^{2}$/Hz and the PSD at Δf is −47.0 dB V${}^{2}$/Hz, giving $\Delta PSD_{\Delta f}$ = 51.8 dB. The noise floor of the demodulated phase in this case is −54.6 dB rad${}^{2}$/Hz, which is much higher than that in channel 163.

### 3.2. Relation between PSD and Noise Floor

The blue line in Figure 6a plots the noise floor of the demodulated phase signal over position. The same dataset as shown in Figure 3 was used. Meanwhile, the relative PSD at the heterodyne frequency is also plotted in the same figure (red line). It is clear that the relative PSD at heterodyne frequency is anti-correlated with the noise floor of the detected phase. The estimated cross-correlation coefficient is ∼−0.99.

To provide more insight on the observation, the blue curve in Figure 7 plots the signal-to-noise ratio (SNR) of the collected intensity signal at different positions. Here, SNR is estimated as:(2)$$SNR=\frac{P_{s}}{P_{n}}=\frac{\int_{f_{s1}}^{f_{s2}}PSD}{\int_{f_{n1}}^{f_{n2}}PSD}$$
where $P_{s}$ and $P_{n}$ are the power level of signal and noise, respectively, PSD represents the power spectrum density of the intensity signal. $\left[f_{s1},f_{s2}\right]$ and $\left[f_{n1},f_{n2}\right]$ are the considered frequency range of signal and noise, respectively. In the calculation, the bandwidth of signal is [49.9kHz,50.1kHz] with the rest considered as noise. As can be seen from Figure 7, the relative PSD at the heterodyne frequency is strongly linked with the SNR (with the estimated cross-correlation coefficient of ∼0.98). This is reasonable since the relative PSD at heterodyne frequency is the difference between the PSD at Δf and the noise floor. In Gabai’s work [22], it has been found that the SNR of the demodulated phase was directly proportional to the SNR of the backscattered trace. According to Equation (Equation 2), the higher the noise floor, the lower the SNR of the demodulated phase, explaining the negative relation between the relative PSD at heterodyne frequency and the noise floor.

Figure 6c shows the RMS value, i.e., the effective amplitude, of the detected phase change over position. It can be seen that, in certain channels, the RMS increases sharply due to the phase fading noises. Figure 8 displays the RMS of the detected phase versus the relative PSD at heterodyne frequency. These data were calculated from a 10-s continuously measured signal. It can be seen that the RMS value increases sharply when the relative PSD at heterodyne frequency is lower than 50 dB (red dashed line).

### 3.3. Weighted-Channel Stack Algorithm

Based on the above observations, we propose a weighted-channel stack algorithm (WCSA) to alleviate the phase fading effect in the HD-DAS system. The procedure of the algorithm is listed as follows:Convert the intensity data collected by DAQ I(t) into parallel data I(z,τ), where *t* and τ refer to the fast-time and slow-time frame [20]. The position *z* corresponds to a specific sampling channel.Divide I(z,τ) into blocks with time interval of 0.1 s for further processing. Then, we have $I\left(z,\tau \right)=\{I\left(z,\tau_{n}\right)|\tau_{n}=\left(\left(n-1\right)*0.1s,n*0.1s\right],n=1,2,\cdots \}$. Note that we used the graphics processing unit (GPU) of the computer for parallel processing of the demodulation algorithm, in which case a 0.1 s time interval was found to be optimal for data processing.Calculate the PSD of I(z,τ) at different channels of each time block; furthermore, find the PSD of heterodyne frequency $PSD_{\Delta f}^{n}\left[I(z,\tau_{n})\right]$ and its noise floor $NF^{n}\left[I(z,\tau_{n})\right]$ within each time block. Calculate the relative PSD at heterodyne frequency of different channels using Equation (Equation 3):
(3)$$\Delta PSD_{\Delta f}^{n}\left(z\right)=PSD_{\Delta f}^{n}\left[I(z,\tau_{n})\right]-NF^{n}\left[I(z,\tau_{n})\right]$$Define the weight of different channels using Equation (Equation 4):
(4)$$w^{n}\left(z\right)=\left\{\begin{array}{ccc} \; & \Delta PSD_{\Delta f}^{n}\left(z\right), & \Delta PSD_{\Delta f}^{n}\left(z\right)\geq 50dB\\ \; & 0, & \Delta PSD_{\Delta f}^{n}\left(z\right)<50dB \end{array}\right.$$The weighted-channel stacked signal can be obtained by Equation (Equation 5), where *M* is the number of channels used for stacking. Note that a larger value of *M* may reduce the spatial resolution of the sensor due to the average effect:
(5)$$I\left(z,\tau_{n}\right)=\sum_{k=z-M}^{z}w^{n}\left(k\right)I\left(k,\tau_{n}\right)$$With the heterodyne demodulation algorithm proposed in [19], the demodulated phase signal can be retrieved.

We then follow the aforementioned procedure to re-demodulate the signal in Figure 3. The result is given in Figure 9. Here, the number of channels for stacking is *M* = 4, which is the optimal stacking number for our data set. Figure 9a shows the RMS of the phase signal over position and the standard deviation of the RMS is 0.15 rad, achieving a noise reduction of 13.7 dB compared to the results in Figure 3 (also see Figure 6b,c for a direct comparison over position). Note that the increase of noise in some channels (e.g., 220 m) may be induced by the noise in adjacent channels after applying WSCA; the overall effect of noise suppression, however, is clear.

### 3.4. Sound Detection

#### 3.4.1. Experimental Setup

As an application, we used our improved HD-DAS system for sound detection. Figure 10a,b illustrate respectively the schematic and physical diagrams of the experimental configuration. At a position of 50 m, a segment of 5 m fiber was set close to a noise source. The piece of the fiber that used to ’listen’ to the music was fixed to the clamps by adhesive tape. The distance between two clamps was approximately 30 cm and the fiber locates at position of about 92 m. A music player was placed underneath the fiber to play the music.

#### 3.4.2. Sound Detection Results

Figure 11 displays the waterfall plot of the retrieved phase signal from the improved HD-DAS system (thus with WCSA algorithm applied). Two sets of vibration signals are visible in the figure: one is located at position 50 m, which is the noise-induced phase change; the other at 92 m represents the music-induced phase change. The gauge length can be calculated as $(w+L_{d})/2$, which is 8 m in this case. This is in accordance with the length of signal span in the waterfall plot. Note that the noise source is set far away from the music player and whose effect on the music-induced phase signal is negligible.

Figure 12 plots the retrieved noise signal located at position 50 m. It can be seen that the noise source is mainly the 50 Hz electronic frequency and its harmonics. The noise at high frequency can be considered as white noise.

Figure 13 shows the retrieved music signals lasted for 205 s with and without WCSA. When WCSA is not used, the phase fading effect exists that can be clearly identified from the demodulated signal at 85 s and 177 s, where the music signals are submerged in the sharply increased phase noises. The retrieved music without WCSA can refer to the Appendix A and the noise interruption can be identified from the audio. The retrieved sound with WCSA is also provided in Appendix A and, in this case, smooth and beautiful music without distortion is obtained. As described previously, the phase fading noise will change with time, while, with the proposed algorithm, the 205 s music signal can be retrieved continuously without distortion, validating the fading alleviation effect of this algorithm.

As a direct comparison, in Figure 14a, we plot the retrieved sound signal from time 85 s to 87 s without (red) and with (blue) WCSA applied, overlapped with the original waveform (black). As can be seen, without WCSA, the retrieved signal shows fatal distortions due to a phase fading effect. Under this circumstance, the PSD of the retrieved phase (shown in Figure 14b) appears significant deviation from that of the original data (especially at low frequency range). On the other hand, when WCSA is used, the PSD of the retrieved phase coincides very well with that of the applied sound.

## 4. Conclusions

In conclusion, we report the experimental observation of an anti-correlation relation between the relative PSD at heterodyne frequency and the noise floor of the detected phase signal in the HD-DAS system. A novel WCSA algorithm was further proposed to suppress the phase fading noise, with the key to enhance the relative PSD at heterodyne frequency. An optimal reduction of noise level of 13.7 dB can be achieved using our approach. We have also demonstrated the improved performance of the system on sound detection. The improved HD-DAS system offers a promising solution to seismic wave detection in down-hole and oceanic environments.

## Figures and Tables

**Figure 1 sensors-20-00999-f001:**
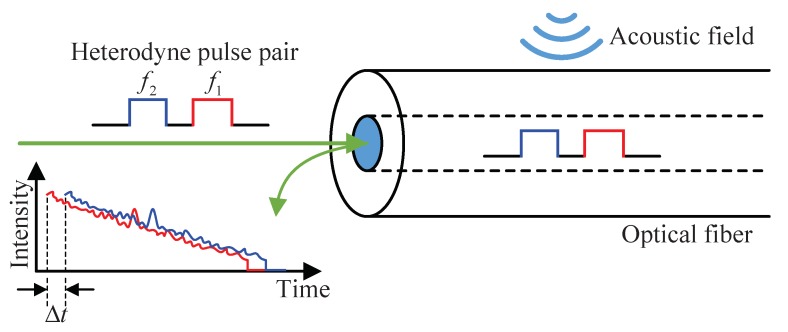
Working principle of the heterodyne demodulated distributed acoustic sensor (HD-DAS) system.

**Figure 2 sensors-20-00999-f002:**
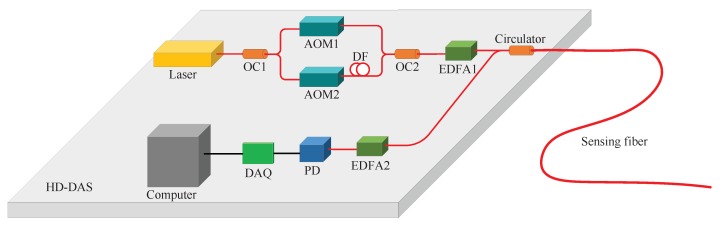
Experimental configuration of HD-DAS system. OC, optical coupler; AOM, acousto-optic modulator; DF, delay fiber; EDFA, Erbium-doped fiber amplifier; PD, photo detector; DAQ, data acquisition card.

**Figure 3 sensors-20-00999-f003:**
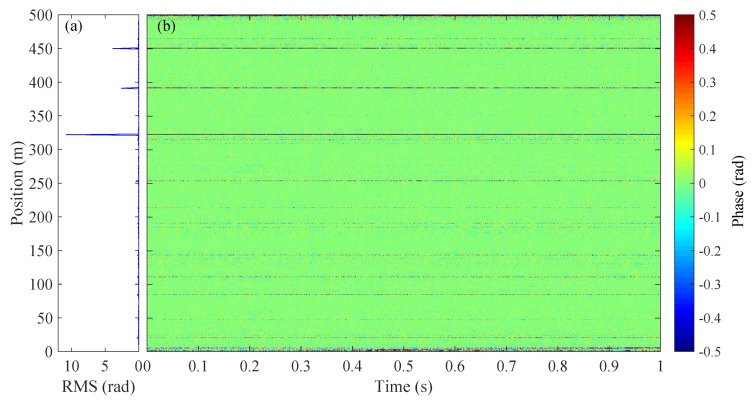
(**a**) RMS and (**b**) waterfall plot of the demodulated phase signal over position.

**Figure 4 sensors-20-00999-f004:**
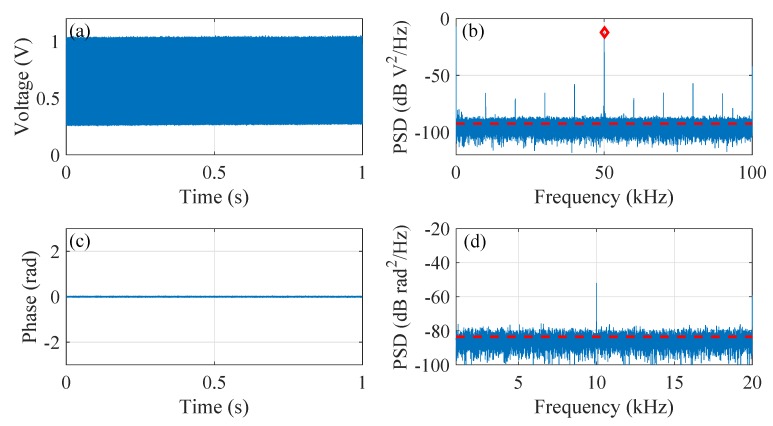
Results for channel 163 (low noise channel). (**a**) the detected intensity signal; (**b**) the PSD of the intensity signal; (**c**) the demodulated phase signal; (**d**) the PSD of the phase signal.

**Figure 5 sensors-20-00999-f005:**
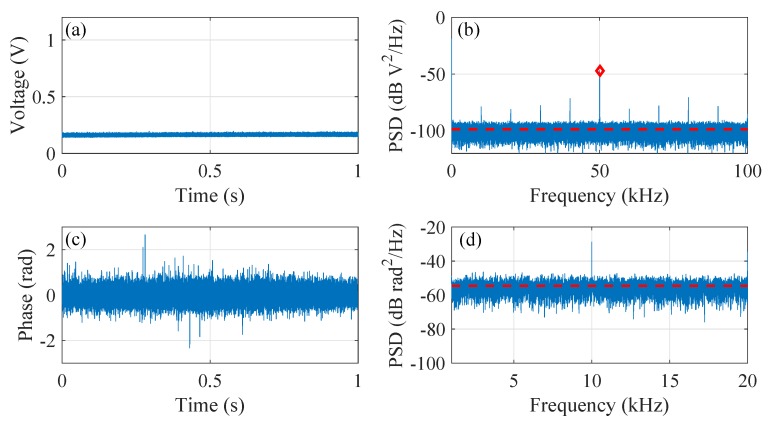
Results for channel 314 (high noise channel). (**a**) the detected intensity signal; (**b**) the PSD of the intensity signal; (**c**) the demodulated phase signal; (**d**) the PSD of the phase signal.

**Figure 6 sensors-20-00999-f006:**
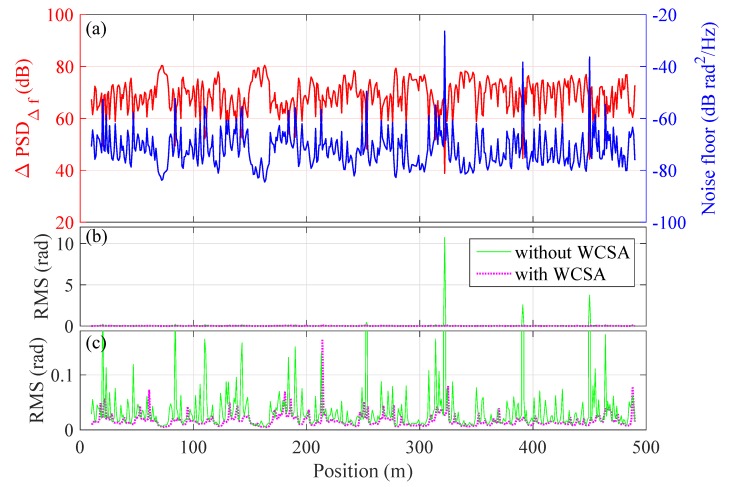
(**a**) the relative PSD at heterodyne frequency (left vertical axis) and the corresponding noise floor of the detected phase (right vertical axis) at different positions, showing a clear anti-correlation relation; (**b**) the RMS value of the detected phase over position without (green-solid) and with (magenta-dashed) WCSA; (**c**) zoom-in of (**b**) in vertical axis.

**Figure 7 sensors-20-00999-f007:**
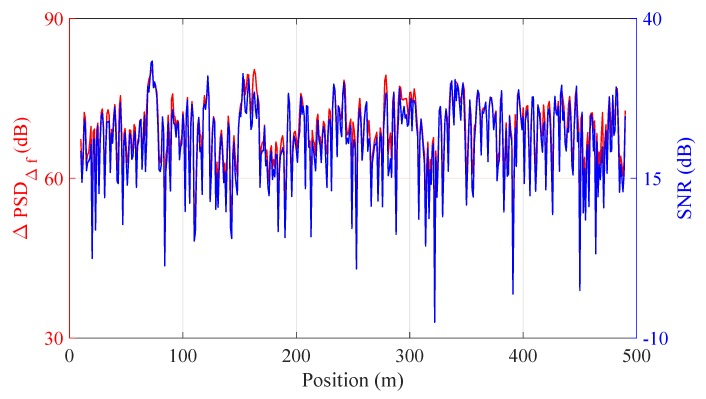
Measured relative PSD at heterodyne frequency (left vertical axis) and the SNR (right vertical axis) over position.

**Figure 8 sensors-20-00999-f008:**
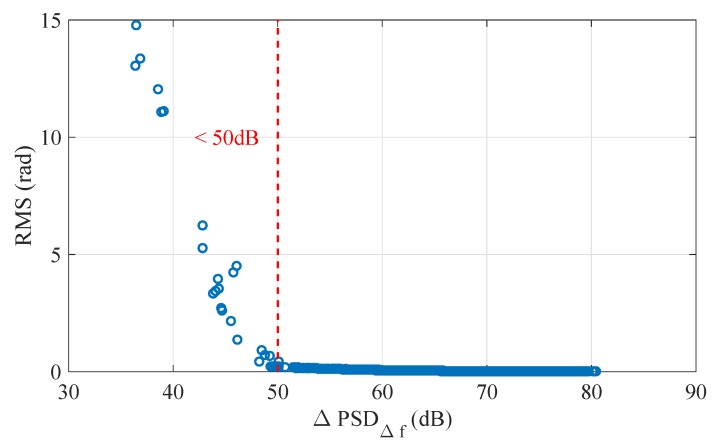
The RMS of the detected phase versus the relative PSD at heterodyne frequency.

**Figure 9 sensors-20-00999-f009:**
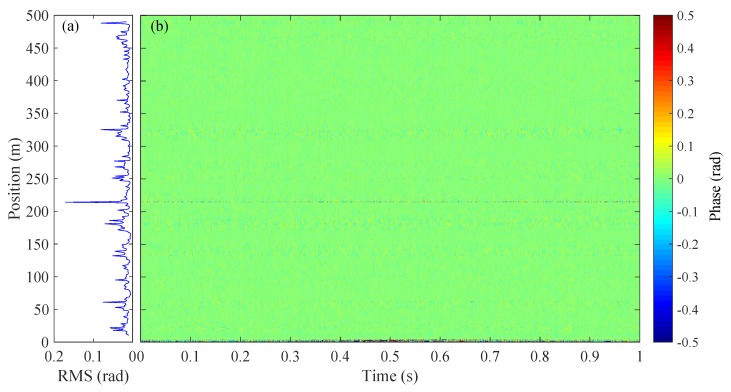
(**a**) RMS and (**b**) waterfall plot of the demodulated phase over position with WCSA applied.

**Figure 10 sensors-20-00999-f010:**
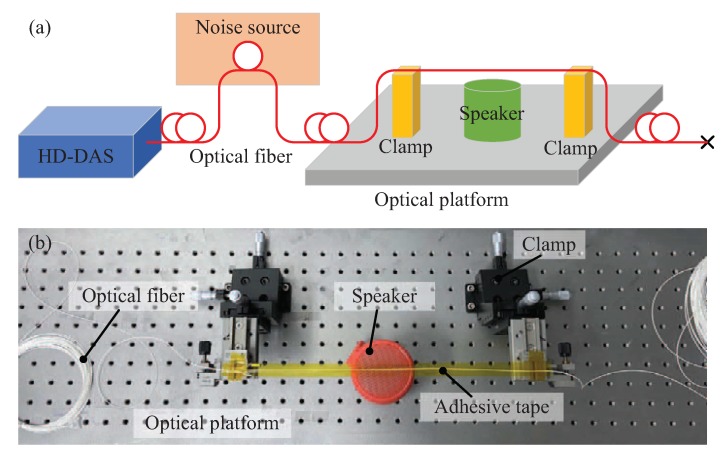
The experimental configuration using the HD-DAS system for sound detection. (**a**) schematic diagram; (**b**) real system in the lab.

**Figure 11 sensors-20-00999-f011:**
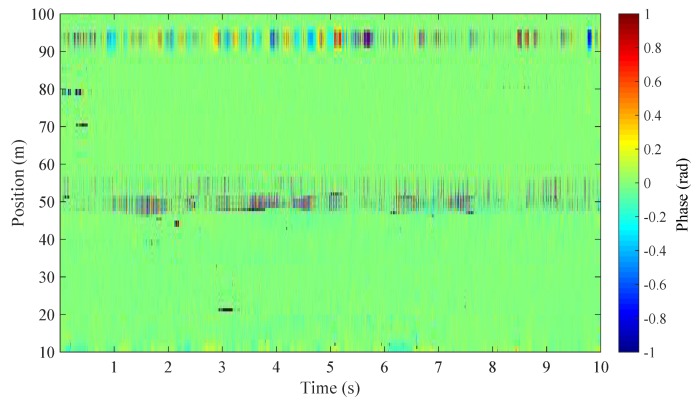
Waterfall plot of the detected phase signal for the music.

**Figure 12 sensors-20-00999-f012:**
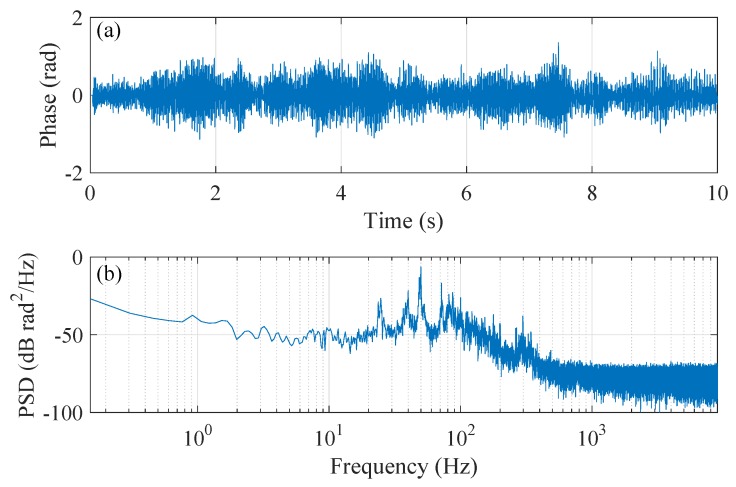
Plot of (**a**) the demodulated phase signal located at position 50 m and (**b**) its PSD.

**Figure 13 sensors-20-00999-f013:**
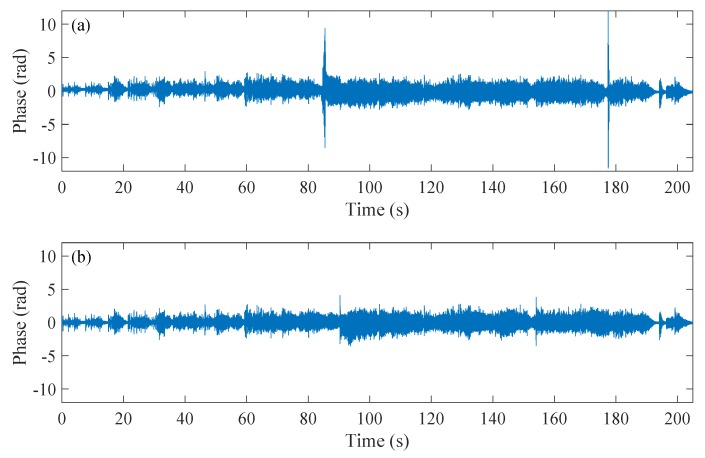
The plot of the retrieved sound signals. (**a**) the retrieved phase signal without WCSA; (**b**) the retrieved phase signal with WCSA.

**Figure 14 sensors-20-00999-f014:**
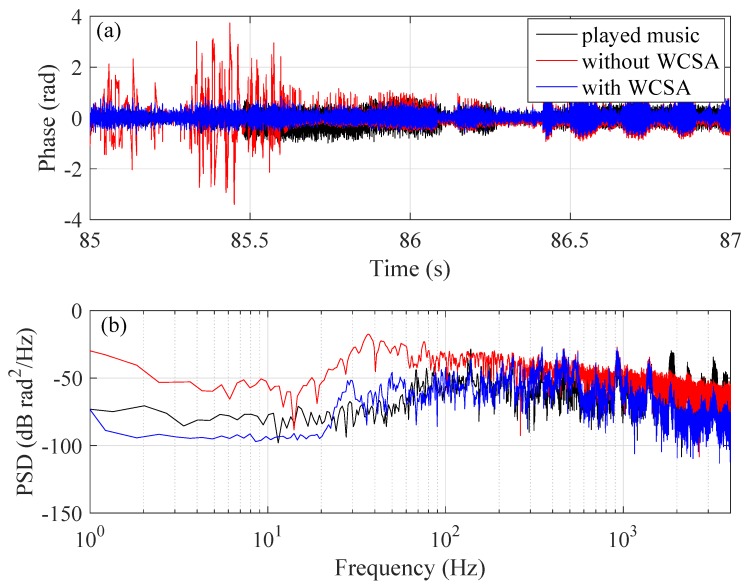
Comparison of the retrieved sound signals without (red) and with (blue) WCSA applied. The black curve is the original music data. (**a**) time-domain signals; (**b**) the PSD of the signals.

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
