# Peer review of "Performance Improvement of Dual-Pulse Heterodyne Distributed Acoustic Sensor for Sound Detection"

_sensors, 2020, doi:10.3390/s20040999_

Round 1

Reviewer 1 Report

Manuscript No:  sensors-693632

Title:  Performance Improvement of Distributed Acoustic Sensor for Sound Detection

Authors:  Xiangge He, Min Zhang, Lijuan Gu, Shangran Xie, Fei Liu and Hailong Lu

Overview In this manuscript the authors report on the theoretical and simulation results on improving the heterodyne demodulated-distributed acoustic sensor system by alleviating the phase fading effect. The contents are expressed clearly; the manuscript is well organized and written in reasonable English. Look for typos. The authors have acknowledged recent related research. As long as my knowledge, the work presented is original and it is correct from a scientific point of view.

Detailed analysis

Abstract: Must be clear, objective and self-explanatory. Please state what have you done, how did you do it, the results you got and the novelty of your work. Do not star with with “Phase fading …”. State if this is experimental of simulation.

Introduction: provides an interesting approach to the subject and there are up to date references.

Working Principle of HD-DAS system

- it is clear and objective.

Signal analysis:

- this section deals with experimental results, please re named it to 3. Results and discussion

- figure 7 needs some error bars.

- why the exponential fitting in figure 7? Please make it clear for the readers.

- what kind of information one gets from figure 3 and 8? They are essentially green rectangls.

A comment would be more effective.

- section 4 and 5 should be included as subsections of  section 3.

- “A music player was placed underneath the fiber to play the music ’Never enough’ sung by Loren Allred”.

Glad there are researchers listen to good music! This is unnecessary information.

Overall assessment

The work reported presents reasonable utility for supplementary studies and developments in the field.

In my opinion it may be eventually published after minor revision.

Review Criteria Scope of Journal

Rating: Moderately high

Novelty and Impact

Rating: Medium

Technical Content

Rating: Medium

Presentation Quality

Rating: Medium

Author Response

We thank the reviewer for the careful reading of the manuscript and their comments. We have responded to all points (in red font) and revised our manuscript according to the comments (in red font). We hope the revised manuscript is now suitable for publication in Sensors.

Please see the attachment for detailed information.

Reviewer 2 Report

The authors proposed a noise reduction method with Weighted-channel stack algorithm to improve the phase fading in distributed acoustic sensor. The manuscript is well written and easy to follow. The noise reduction of 13.66dB is good. I suggest publishing with minor changes. Only one comment: The comparison between figure 3 and figure 8 are not obvious. It may be helpful to show data in 1 second, as appose to 10 seconds, so that the details are distinguishable.

Author Response

Thank you for the careful reading of the manuscript and the comments. We have responded to all points (in red font) and revised our manuscript according to the comments (in red font). We hope the revised manuscript is now suitable for publication in Sensors.

Please see the attachment for detailed information.

Reviewer 3 Report

The authors have analyzed the performance of a dual-pulse heterodyne demodulated DAS, and proposed an algorithm to reduce the phase fading noise. I recommend the publication in Sensors as long as the authors address the following comments:

As far as I understand, the proposed WCSA algorithm is only valid in their dual pulse heterodyne configuration. Hence, I believe that the authors should clarify this point in the title, i.e., by specifying that the performance improvement is achieved in a dual pulse heterodyne DAS configuration. The current title can be misleading (it cannot be performed in more traditional DAS using a single probe pulse). In lines 36-37, do the authors mean ‘destructively’? Besides, the entire sentence should be carefully revised. It has several typos, e.g., “superimposes”, “vanishingly”. The fact that the so-called relative PSD at heterodyne frequency (which is kind of a SNR measurement of the interferometric signal at each channel) has a negative relationship with the noise floor of the demodulated phase is not surprising. A statistical analysis of SNR of phase-sensitive OTDR performed by H. Gabai et al. [R1] concluded that the SNR of the demodulated phase is directly proportional to the SNR of the backscattered trace (note that the ‘signal’ in the demodulated phase is proportional to the perturbation, and hence it should be a constant value). This conclusion is general for any phase-demodulated phiOTDR system. The relative PSD at heterodyne frequency calculated by the authors must be directly related with the SNR of the backscattering trace. It will be more interesting if the analysis performed by the authors considers existing related literature (e.g., proving the relationship between the relative PSD and the SNR of the trace, which is a parameter easily understandable by any reader with some knowledge about DAS, and then relate those parameters with the phase-demodulated noise floor as in [R1]).

[R1] H. Gabai and A. Eyal, “On the sensitivity of distributed acoustic sensing,” Opt. Lett, 41, 5648 (2016)

In DAS systems, laser phase noise is a limiting factor of performance, critically affecting the trace SNR and consequently the operation range and the sensitivity. In this work, authors employ an ultra-narrow width laser, minimizing the laser phase noise. Still, in the discussion on phase noise in Section 3, there is no reference to laser phase noise. Similarly to laser intensity noise, I understand that its contribution will be identical to all sample channels. How laser phase noise would affect the measurement in the dual pulse heterodyne configuration? Would the WCSA algorithm be still useful in a system limited by laser phase noise (i.e., a system using a broader linewidth laser)? Could the author establish the conditions of utility of the WCSA algorithm? Why there is a peak at 10 kHz in the power spectra of the demodulated phase (Fig. 4 (d) and 5 (d))? If the fiber is unperturbed, I would have expected to see simply noise in the PSD of the phase. The second point of the WCSA algorithm is not clear to me. Why the matrix I(z,τ) has to be divided in blocks of 0.1 s. Are those channels averaged? Why 0.1 s, is it an arbitrary value or it has been chosen following any criterion? In general, the orthography must be carefully revised. There are many errors, especially in verbs conjugation in present tense (lack of ‘s’ in the third person), use of gerund after prepositions, etc.

Author Response

(The authors gave the same response as above.)

Reviewer 4 Report

The investigations presented in this manuscript build upon previous work of most of the same authors, namely a distributed acoustic sensing (DAS) scheme based on heterodyne demodulation. In the present paper the authors describe a method of measurement data pre-processing which they claim improves the performance of their demodulation scheme with respect to phase noise suppression and subsequent distortion-free sound detection. The described weighted-channel stack algorithm (WCSA) is designed by the authors following the results from statistical and spectral analyses of phase noise characteristics from a perturbation-free (without external signal) experimental measurement. The authors aim to demonstrate the benefits of the algorithm for recovery of a music piece played during an experiment.

The paper is well-structured and the topic is, in general, of interest to readers. Nevertheless, the manuscript suffers from quite a few shortcomings. These relate to the presentation of data, insufficient description of experiments, partially unclear explanation of the data processing, imprecise phrasing and, foremost, missing reasoning behind chosen approaches. The authors do not offer an explanation or conjecture about the reason for their main results of the phase noise analysis, that the PSD at heterodyne frequency PSD_(Delta f) is strongly anticorrelated with the phase noise, even though this behavior seems intuitive, given the principle of their modulation scheme. To demonstrate the claimed improvement of retrieved signal quality due to their processing scheme, the authors present insufficient evidence and unsuitable graphical representations, no quantifyable measures/indicators for signal improvement are shown. Instead, two versions of retrieved sounds (using WCSA and not using it) are described and given as supplementary material. However, adequate evidence for claimed improvement is paramount and must be presented in the body of the manuscript [on a sidenote: I was unable to access the supplementary material during my review. That does, however, not affect my rating of the manuscript.]. Furthermore, there are minor errors/ inconsistencies referring to specific given values in the manuscript as well.

This manuscript should thus only be reconsidered for publication in Sensors after major revisions which address the above mentioned problems. Alternatively, the authors could also rewrite the paper and resubmit. In the following, I list specific corrections/additions and questions that should be addressed by the authors:

Abstract, line 3: please refrain fom phrasings such as "has a good negative correlation with", use e.g. "is strongly anti-correlated with". Abstract, line 8: phrasing "for the first time" unclear, please remove Abstract (and last sentence Conclusions): The claimed suitability of the improved HD-DAS scheme specifically for geological exploration is addressed nowhere else in the manuscript. Why is the presented scheme specifically well-suitable for geological explorations? These kinds of investigations focus on comparably low-frequency, low-amplitude signals, while in their presented manuscript, wideband (music) signals are processed. Please expand on this in the Conclusions/Discussion or generalize the potential applications. Section 1, line 40: "However, this scheme..." - sentence unclear, please rephrase Section 1, line 54: "good negative relationship" - imprecise, please rephrase (see above) all over manuscript: for given values consider only significant digits; most results are statistical/ depend on data specific data subset; thus something like Delta f=-46.99 dB V^2/Hz or "noise reduction of 13.66 dB" does not make sense Section 2, line 88: is the (very) narrow linewidth of the interrogator laser of 100 Hz necessary for the HD-DAS scheme to work? If so, please elaborate. Section 3, first paragraph: the inhomogeneity of the phase noise is attributed to fading but the given peculiaties of fading are somewhat imprecise or unclear. Fading can in general occur in all channels, depending on environmental conditions, sensor application etc, while it is true, that certain sensing positions (channels) are more prone to fading over long periods, fading is a time-dependent phenomenon. readers could get the impression from the current phrasing that fading only takes place in certain channels and never in the others. Fig. 3 and all other relevant Figs: please change color bar label/ axis label from "amplitude (rad)" to "phase (rad)", as "amplitude" is confusing Figs. 4&5: for more clarity and easier comparison between the characteristics in the noisy and non-noisy channel, please adjust Fig.4c) and Fig. 5c) to the same scale; add "low noise channel" etc. or similar to each caption. Figs. 4&5 a) show only a detail of the entire time traces, I assume the PSD is nonetheless calculated for the entire 10s signals? How is the noise floor determined/calculated? Please add to text.  Figs. 4&5 b) what is the origin of the peaks at 10 kHz and harmonics? Not mentioned in the text. Section 3, line 118: given NF value does not correspond to the one given in Fig. 4d) Section 3, line121: "only" - the given value is actually higher than the one given for channel 163 Section 3, line 122: given NF value does not correspond to the one given in Fig. 5d) Fig. 6: are the measures calculated from the dataset as shown in Fig. 3? Section 3, lines 123-130 & Fig. 6: strong correlation is not explained or conjectured about; RMS is only "effective amplitude" of the phase change, if time-invariant fading within the considered time interval is assumed. This is reflected in the data shown in Fig. 6b): only the channels which are constantly faded exhibit pronounced peaks, temporily strongly faded channels have peaks that can be barely made out. In that sense, what is the additional insight, Fig. 6b) and the RMS measure calculated over the entire 10s measurement give? Section 3, Fig. 7 and line 133: exponential fit and fit formula are redundant if no underlying model is used/mentioned Fig. 7: Where does the RMS data come from? It is not (or not completely) shown in Fig. 6b). Especially the high RMS values at PSD values <40 dB are not shown. Section 4, description of WCSA: the subdivisions into 0.1s blocks are not explicitely reflected in equations 2-4 Eqs. 2&3: Delta PDS_delta f does not show a time-dependence, only a spatial one. Correspondingly the weight w is also shown as single scalar per channel. Please correct to reflect time-variation of fading and use of 0.1s blocks. Eq. 4: this implies an increasing spatial uncertainty with increasing averaging size M, please mention in text Section 4, line 153: std of RMS of 0.7223 rad for unprocessed phase signal is not mentioned previously in the text nor is it clear for which time interval or pos. channel it is calculated. Is it an overall value for the entire data sets shown in Figs. 3 and 8, respectively? Fig. 8: Why was M=4 chosen, better noise suppression for M>4? Also, it seems previously non-noisy channel around pos.~220 m now exhibits strong fading after processing with WSCA. How? Also, how does a corresponding NF plot like Fig. 6a) look like for the case where WSCA was applied? Section 5, Fig. 9a) the fiber segment around pos. 50m close to the noise source is not depicted in the diagram Section 5, desciption experiment and Fig. 10: what are the characteristics of the noise? White noise, colored noise, amplitude relative to music amplitde? Section 5, Fig. 11 & lines 170-178: Fig. 11 alone is supposed to demonstrate the improved performance of the scheme with WSCA applied. That is completely insufficient. Except for the two mentioned signal peaks in Fig. 11a) basically no qualitative difference between the depicted traces is discernible with the naked eye. The so-claimed suppressed distortions can of course not be seen. Is the shown signals extracted for pos. 92 m exactly? Please add some additional quantifiable results that can serve as a measure for the improvement of the signal quality. For that, a direct comparison of the raw input waveform of the digitized music with the retrieved waveform for both cases would help. This could be done in terms of correlation, error rate, signal overlay, spectral analysis etc. Also, measures comparing harmonic distortion, which is a common problem using phi-OTDR could be useful. A reference to the supplementary material (wav files) alone to gauge the benefits of the presented algorithm is certainly a nice-to-have but in no way sufficient evidence. The reader must be able to access the results and evaluate the evidence for claimed performance improvements from reading the paper alone. Please adapt and expand this part of the manuscript. This missing accessible quantifiable evidence is the most serious problem in the manuscript. Without it, the present work should not be published.

Author Response

(The authors gave the same response as above.)

Round 2

Reviewer 3 Report

I recommend the publication in Sensors of the revised manuscript in its current form. I suggest to take into consideration the following comments, to be implemented at the author’s discretion.

The newly introduced paragraphs sometimes are not well integrated within the original text, making the reading a bit difficult. For example, the new explanation at the beginning of page 7 (below Fig. 7) is introduced quite abruptly. I suggest the authors to take care of the readability of the text, for example, by adding some introductory sentence to help the reader to better follow the reasoning.  

In line 145, when the authors refer Eq. (3), the expression is too far from this first reference, even in a different section. The authors should reconsider the position of the equation.

The authors comment several times (at least twice) that certain mathematical derivations will be published elsewhere. In my opinion, mathematical derivations important to understand the present work should be included in this paper, either as an annex or as supplementary material. Besides, the fact that this comment is done several times does not give a good impression of the quality of the present paper, that seems like a fraction of the whole piece of work. If the authors are convinced that those mathematical derivations merit a different publication, perhaps they should not be so explicit in this manuscript.

Author Response

Thank you for the careful reading of the manuscript and the comments. We have responded to all points (in red font) and revised our manuscript according to the comments (in red font).

Reviewer 4 Report

Previous comments and concerns have been adequately addressed by the authors in the revised version of the manuscript. I support publication in Sensors after the following few minor changes have been implemented:

The authors have changed their phrasing regarding the anti-correlation to "negative correlation" (Abstract, Introduction) which is fine, the magnitude of correlation, however, becomes only clear after reading the main paper. I suggest more emphasis, like "significant" or "strong negative correlation", since this is one of the most important points. To avoid confusion between the PSD, relative PSD and such values, please add the symbol "\Delta PSD_{Delta f}" to the specific values for the relative PSD in lines 122 and 129. Please add to the first sentence p. 5 such as to reflect also the peaks at 10kHz and harmonic frequencies in Fig. 4b) At the authors discretion, I suggest improving the text flow especially page between end of page 6 and first paragraph p.7, and second paragraph p.7, the newly introduced text somewhat reduces readability the referencing of a future different publication about the signal analytics (p. 7) is unnecessary, reference to manuscript length/scope is sufficient. Similarly for the reasoning behind M=4 on p. 8. The use of a specific value for a data processing parameter which has been found to be optimal for the given data set is legitimate (delete text in parenthesis in lines 171-172).

Author Response

(The authors gave the same response as above.)
